# EXPLAINING TIME SERIES BY COUNTERFACTUALS

## ABSTRACT

We propose a method to automatically compute the importance of features at every observation in time series, by simulating counterfactual trajectories given previous observations. We define the importance of each observation as the change in the model output caused by replacing the observation with a generated one. Our method can be applied to arbitrarily complex time series models. We compare the generated feature importance to existing methods like sensitivity analyses, feature occlusion, and other explanation baselines to show that our approach generates more precise explanations and is less sensitive to noise in the input signals.

## 1 INTRODUCTION

Multi-variate time series data are ubiquitous in application domains such as healthcare, finance, and others. In such high stakes applications, explaining the model outcome is crucial to build trust among end–users. Finding the features that drive the output of time series models is a challenging task due to complex non-linear temporal dependencies and cross-correlations in the data. The explainability problem is significantly exacerbated when more complex models are used. Most of the current work in time series settings focus on evaluating globally relevant features (Yang et al., 2005; Yoon et al., 2005; Hmamouche et al., 2017). However, often global feature importance represents relevant features for the entire population, that may not characterize local explanations for individual samples. Therefore we focus our work on individualized feature importance in time series settings. In addition, besides identifying relevant features, we also identify the relevant time instances for those specific features, i.e., we identify the most relevant observations. To the best of our knowledge this is the first sample–specific feature importance explanation benchmark at observation level for time series models.

In this work, we propose a counterfactual based method to learn the importance of every observation in a multivariate time series model. We assign importance by evaluating the expected change in model prediction *had an observation been different*. We generate plausible counterfactual observations based on signal history, to asses temporal changes in the underlying dynamics. The choice of the counterfactual distribution affects the quality of the explanation. By generating counterfactuals based on signal history, we ensure samples are realistic under individual dynamics, giving explanations that are more reliable compared to other ad-hoc counterfactual methods.

## 2 METHOD: FEED-FORWARD COUNTERFACTUALS FOR TIME SERIES EXPLANATION

In this section we describe our method, Feed Forward Counterfactual (FFC), for generating explanation for time series models. A feature is considered important if it affects the model output the most. In time series, the dynamics of the features also change over time, which may impact model outcome. As such it is critical to also identify the precise time points of such changes.

In order to find important observations for high-dimensional time series models, we propose to use a counterfactual based method. Specifically,

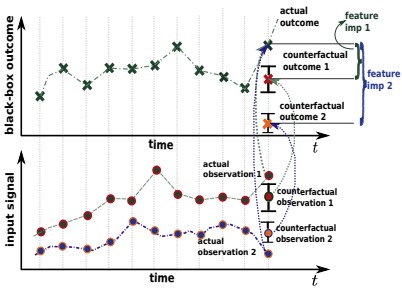

Figure 1: Illustration of Proposed Method

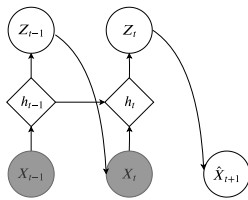

(a) Graphical model of the conditional generator. $h_t$ represents the hidden state of an RNN cell at time $t$, and $z_t$ is the latent representation of the history

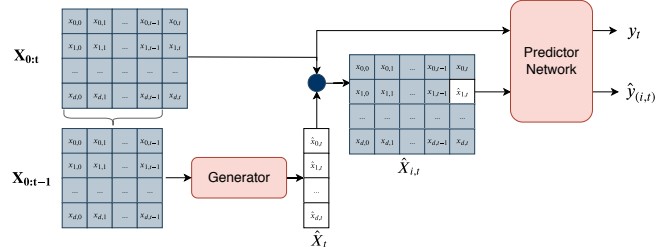

(b) FFC procedure: Counterfactual $\hat{X}_t$ is generated using signal history. We look into the difference of the original output $y_t$ and $\hat{y}_{(i,t)}$ where observation $x_{(i,t)}$ is replaced with a counterfactual.

Figure 2: Components describing the proposed method.

we assign importance to an observation $x_{i,t}$ (feature $i$ at time $t$) based on its effect on the model output at time $T(> t)$. We replace observation $x_{i,t}$ with a counterfactual $\hat{x}_{i,t}$, to evaluate this effect. Figure 1 demonstrates how importance of a feature is characterized by replacing an observation with a counterfactual.

## 2.1 NOTATION

Multi-variate time series data is available in the form of $\mathbf{X}^{(n)} \in \mathbb{R}^{d \times T}$ (where $d$ is the number of features with $T$ observations over time) for $n \in [N]$ samples. We are interested in black-box estimators that receive observations up to time $t$, $\mathbf{x}_t \in \mathbb{R}^d$, and generate output $y_t$ at every time point $t \in [T]$. $\mathcal{F}$ denotes the target black-box model we wish to explain through the proposed approach called FFC. For exposition, throughout the paper, the index $n$ over samples has been dropped for notational clarity. We index features with subscript $i$. $\mathbf{x}_{-i,t}$ indicates features at time $t$ excluding feature $i$. The notation used for exposition work is briefly summarized in Table 1.

| Notation | Description |
|---|---|
| $[K]$ for integer $K$ | Set of indices $[K] = \{1, 2, \ldots, K\}$. |
| $i, t, n$ | Index for feature $i$ in $[d]$, time step $t$ and sample $n$ in $[N]$ respectively |
| $-i$ | Set $[d] \setminus i$ |
| **Observations and Outcomes** | |
| $x_{i,t}$ | Observation $i$ at time $t$. |
| $\mathbf{x}_t \in \mathbb{R}^d$ | Vector $[x_{1,t}, x_{2,t}, \cdots, x_{d,t}]$ |
| $\mathbf{X}_{0:t} \in \mathbb{R}^{d \times t}$ | Matrix $[\mathbf{x}_0, \mathbf{x}_1, \cdots, \mathbf{x}_t]$ |
| $y_t \triangleq \mathcal{F}(\mathbf{X}_{0:t})$ | Observed outcome of the black-box model $\mathcal{F}$, at time $t$ |
| **Generative Model and Estimator** | |
| $\mathcal{G}_i : \mathbb{R}^{d-1 \times m} \to \mathbb{R}$ | Conditional Generative Model sampling for feature $i$ |
| $z_t \in \mathbb{R}^m$ | Latent encoding of history up to $t$ |

Table 1: Notation used in the paper.

## 2.2 EXPOSITION

We assign importance score to each observation $x_{i,t}$ for $t \in [T]$ and $i \in [d]$, following the definition:

**Definition 1. Feature Importance:** The importance of the observation $i$ at time $t$, denoted by $Imp(i,t)$ is defined as $E_{p(\hat{x}_{i,t}|\mathbf{X}_{0:t-1})}[|\mathcal{F}(\mathbf{X}_{0:t}) - \mathcal{F}(\mathbf{X}_{0:t-1}, \mathbf{x}_{-i,t}, \hat{x}_{i,t})|]$, where $|\cdot|$ denotes the absolute value and $\hat{x}_{i,t}$ is the counterfactual sample.

That is, the importance of an observation for feature $i$ at time $t$ is defined as the change in model output when the observation is replaced by a generated counterfactual. The counterfactual observation can come from any distribution, however the quality of the counterfactual random variable directly affects the quality of the explanation. We generate the counterfactual sample conditioned on signal history up to time $t$ by sampling from the distribution $p(x_{i,t}|\mathbf{X}_{0:t-1})$. Using a conditional generator

guarantees that our counterfactuals are sampled not only within domain but also specifically likely under the individual sample $\mathbf{X}^{(n)}$, as opposed to having a generator that models data across population. Conditioning on the history also allows us to learn the dynamics of the signal and therefore generate a plausible counterfactual given the past observations.

$p(\mathbf{x}_t|\mathbf{X}_{0:t-1})$ represents the distribution at time $t$, if there were no change in the dynamics of the signals. The counterfactual $\hat{x}_{i,t}$ is sampled from the marginal distribution $p(bx_{i,t}|\mathbf{X}_{0:t-1})$, obtained from $p(\mathbf{x}_t|\mathbf{X}_{0:t-1})$. Let $\mathcal{F}(\mathbf{X}_{0:t-1}, \mathbf{x}_{-i,t}, \hat{x}_{i,t})$ be the output of the model at time $T$, when $x_{i,t}$ is replaced by the generated counterfactual $\hat{x}_{i,t}$. We estimate feature importance $Imp(i,t)$ as $E_{p(\hat{x}_{i,t}|\mathbf{X}_{0:t-1})}[|\mathcal{F}(\mathbf{X}_{0:t}) - \mathcal{F}(\mathbf{X}_{0:t-1}, \mathbf{x}_{-i,t}, \hat{x}_{i,t})|]$, summarized in figure 2(b).

## 2.3 PROPERTIES

Our proposed method has the following compelling properties in explaining the estimator $\mathcal{F}$:

**Time Importance (TI)** For every time series, highlighting relevant time events for different features is important for actionability of the explanations. For instance in a clinical setting, just knowing a feature like heart rate is relevant, is not sufficient to intervene - it is also important to know when a deterioration had happened. With FFC, the most *eventful* time instances can be obtained as:

$$\arg\max_{t\in[T]}\{Imp(i,t) \, \forall i \in [d]\} \qquad (1)$$

We can thus rank time instances in order of importance. That is, time $t_1 \preccurlyeq t_2$, if $\max_{i\in[d]}\{Imp(i,t_1)\} \geq \max_{i\in[d]}\{Imp(i,t_2)\}$.

**Feature Importance (FI)** At any time instance $t$, our method assigns importance to every feature of $x_{i,t}$.

The magnitude of our importance function reflects relative importance. Comparing the importance values across features gives the flexibility to report a subset of important features at each time point $t$ and also reflects the correlation between various features of the time series.

## 2.4 GENERATOR MODEL FOR CONDITIONAL DISTRIBUTION $p(\mathbf{x}_t|\mathbf{X}_{0:t-1})$

We approximate the conditional distribution of $p(\mathbf{x}_t|\mathbf{X}_{0:t-1})$ using a recurrent latent variable generator model $\mathcal{G}$, introduced in Chung et al. (2015). The architecture we use is provided in Figure 2(a). The conditional generator $\mathcal{G}$ models $p(\mathbf{x}_t|z_{t-1})$ where $z_{t-1} \in \mathbb{R}^k$ is the latent representation of history of the time series up to time $t$. The latent representation is a continuous random variable, modeling the distribution parameters. We only use past information in the time series to reflect temporal dependencies. Using the recurrent structure allows us to model a non-stationary generative model that can also handle varying length of observations. Implementation details of the generator are in the Appendix.

Our counterfactuals are not derived by looking at future values which could be done for reliable imputation. Counterfactuals should represent the past dynamics. Note that our derived feature importance is limited by the quality of imputation that may have been utilized by the black–box risk predictor. For experimental evaluation on the effect of generator specifications on counterfactuals and the quality of explanations, see Section 4.1.

## 2.5 FEATURE IMPORTANCE ASSIGNMENT ALGORITHM

The proposed procedure is summarized in Algorithm 1. We assume that we are given a trained block box model $\mathcal{F}$, and the data (without labels) it had been trained on. Using the training data, we first train the generator that generates the conditional distribution, (denoted by $\mathcal{G}$). In our implementation we model $\mathbf{x}$ as a multivariate Gaussian with full covariance to model all existing correlation between features. The counterfactual $\hat{x}_{i,t}$ is then sampled from $\mathcal{G}$ and passed to the black-box model to evaluate the effect on the black-box outcome.

## 3 RELATED WORK

A common method of explaining model performance, in time–series deep learning, is via visualization of activations of latent layers (Strobelt et al., 2018; Siddiqui et al., 2019; Ming et al., 2017) or via

---

**Algorithm 1 FFC**

**Input: $\mathcal{F}$: Trained Black-box predictor model, $\mathbf{X}_{0:T}$, where $T$ is the max time and $S$: Number of Monte-Carlo samples**

---

1: Train $\mathcal{G}$
2: **for all** $t \in [T]$ and $i \in [d]$ **do**
3:     $y_T = \mathcal{F}(\mathbf{X}_{0:t})$
4:     $p(\mathbf{x}_t | \mathbf{X}_{0:t}) \sim \mathcal{G}(\mathbf{X}_{0:t-1})$
5:     **for all** $s \in [S]$ **do**
6:         Sample $\hat{x}_{i,t}^{(s)} \sim p(x_{i,t} | \mathbf{X}_{0:t-1})$
7:         $\hat{y}_T^{(s)} = \mathcal{F}(\mathbf{X}_{0:t}, \mathbf{x}_{-i,t}, \hat{x}_{i,t}^{(s)})$
8:         $imp_T^{(S)} = |y_T - \hat{y}_T^{(s)}|$
9:     $Importance\_Matrix(i,t) = \frac{\Sigma_{s=0}^{S} imp_T^{(S)}}{S}$
10: Return $Importance\_Matrix$

---

sensitivity analysis (Bach et al., 2015; Yang et al., 2018). Understanding latent representations, sensitivity and its relationship to overall model behavior is useful for model debugging. However, these but are too refined to be useful to the end users like clinicians.

Attention models (Vaswani et al., 2017; Vinayavekhin et al., 2018; Xu et al., 2018) are the most commonly known explanation mechanisms for sequential data. However, because of the complex mappings to latent space in recurrent models, attention weights cannot be directly attributed to individual observations of the time series (Guo et al., 2018). To resolve this issue to some extent, Choi et al. (2016) propose an attention model for mortality prediction of ICU patients based on clinical visits. However attention weights may not be consistent as explanations (Jain and Wallace, 2019).

In vision, prior works tackle explainability from the counterfactual perspective, finding regions of the image that affect model prediction the most. Fong and Vedaldi (2017) assumes higher importance for inputs that when replaced by an uninformative reference value, maximally change the classifier output. A criticism to such methods is that they may generate out-of-distribution counterfactuals, leading to unreliable explanations. Chang et al. (2019) address this issue for images using conditional generative models for inpainting regions with realistic counterfactuals.

Evaluating sample based feature importance remains largely unstudied for time series models. While more widely studied for image classification, (Bach et al., 2015; Fong and Vedaldi, 2017) these methods cannot be directly extended to time series models due to complex time-series dynamics. Most efforts in this domain focus on population level feature importance (Tyralis and Papacharalampous, 2017). Suresh et al. (2017) is one of the few methods addressing sample based feature importance and use a method similar to Fong and Vedaldi (2017), called "feature occlusion". They replace each time series observation by a sample from uniform noise to evaluate its effect on model outcome to attribute feature importance. We argue that carefully choosing the counterfactual selection policy is necessary for derive reliable importances. Specifically, replacing observations with noisy out-of-domain samples can lead to arbitrary changes to model output that are not reflective of systematic behavior in the domain. Even if an observation is sampled from the domain distribution, it does not characterize temporal dynamics and dependencies well enough, potentially highlighting features that only reflect global model behavior, as opposed to sample specific feature importance. We therefore model the data–distribution in order to generate reliable counterfactuals. We demonstrate the implications of the choice of the generator (and hence the *counterfactuals*) on the quality of explanation.

## 4   EVALUATION

We evaluate our explainability method for finding important features in time series on 2 simulated datasets and 2 real datasets. Our goal is two-fold a) comparison to existing feature importance baselines in time series and b) evaluating the choice of generators on the quality of counterfactuals and explanations.

We compare to existing feature importance baselines described below:

1. **Feature Occlusion (FO) (Suresh et al., 2017)**: Method introduced in Suresh et al. (2017). This method is an ad-hoc approach for generating counterfactuals. When replacing $x_{i,t}$ with a random sample from the uniform distribution, the change in model output defines the importance for $x_{i,t}$.

2. **Augmented feature occlusion (AFO)**: We augment the method introduced in Suresh et al. (2017) by sampling counterfactuals from the bootstrapped distribution over each feature. This avoids generating out-of-distribution samples.

3. **Sensitivity Analysis (SA)**: This method evaluates the sensitivity of the output to every observation, by taking the derivative of $y_t$ with respect to $x_{i,t}$, at every time point.

4. **LIME (Ribeiro et al., 2016)**: One of the most commonly used explainabilty methods that assigns local importance to features. Although LIME does not assign temporal importance, for this baseline, we use LIME at every time point to generate feature importances.

## 4.1 SIMULATED DATA I

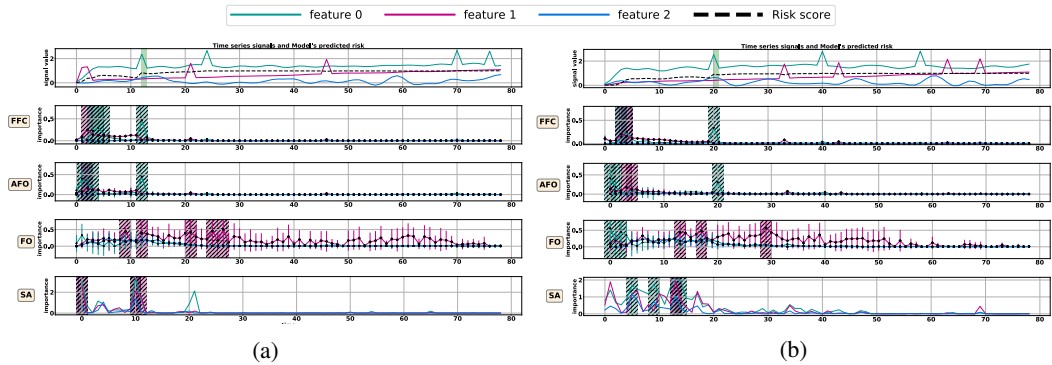

(a)             (b)

Figure 3: Two samples showing performance on simulated data. Top row are the original sampled signals (one per subfigure). FFC and AFO assign importance at the time of spike. Additional samples are in Appendix A.1.

Evaluating the quality of explanations is challenging due to the lack of a gold standard/ground truth for the explanations. Additionally, explanations are reflective of model behavior, therefore such evaluations are tightly linked to the reliability of the model itself. Therefore we created the simulated environment in order to test our method.

In this experiment, we simulate a time series data such that only one feature determines the outcome. Specifically, the outcome (label) changes to 1 as soon as a spike is observed in the relevant feature. We keep the task fairly simple for two main reasons: 1) to ensure that the black-box classifier can indeed learn the right relation between the important feature and the outcome, which allows us to focus on evaluating the quality of the explanations without worrying about the quality of the classifier. 2) to have a gold standard for the explanations since the exact important event predictive of the outcome are known. We expect the explanations to assign importance only to the one relevant feature, at the exact time of spike, even in the presence of spikes in other non-relevant features.

To simulate these data, we generate $d = 3$ (independent) sequences as a standard non–linear auto-regressive moving average (NARMA) time series of the form: $x(t + 1) = 0.5x(t) + 0.5x(t)\sum_{i=0}^{l-1} x(t - l) + 1.5u(t - (l - 1))u(t) + 0.5$ for $t \in [80]$, where the order is 2 and $u \sim \text{Normal}(0, 0.01)$. We add linear trends to the features and introduce random spikes over time for every feature. Note that since spikes are not correlated over time, no of the generators (used in FFC, AFO, FO) will learn to predict it. The important feature in this setting is feature 1. The complete procedure is described in Appendix A.2.1. We train an RNN-based black-box model on this data, resulting in AUC= 0.99 on the test set.

Figure 7 demonstrates explanations of each of the compared approaches on simulated data for 2 test samples. As shown in Figure 7(a), Sensitivity analysis does not pick up on the importance of the spike. Feature occlusion gives false positive importance to spikes that happen in non-important

signals as well as the important one. Augmented feature occlusion resolves this problem since it samples the counterfactuals from the data distribution, however, it generates noisier results as it samples from the bootstrap distribution. The proposed method (FFC) only assigns importance to the first feature at the time of spike. Hence, FFC generates fewer false relevance scores.

Note that all baseline methods provide an importance for evry sample at every time point. The true explanation should highlight feature 1 at time points of spike. Using this ground truth, we evaluate the AUROC and AUPRC of the generated explanations denoted by (exp). Table 2 summarizes the results for simulated data.

|  | Simulated Data I | | Simulated Data II | | |
| --- | --- | --- | --- | --- | --- |
| Method | AUROC (exp) | AUPRC (exp) | AUROC (exp) | AUPRC (exp) | Log-probabilities (counterfactuals) |
| FFC | **0.9995** | **0.8859** | **0.9548 (0.0057)** | **0.2599 (0.0057)** | **-5106586.33** |
| AFO | 0.9901 | 0.3768 | 0.724 (0.012) | 0.0374 (0.0028) | -5134432.65 |
| FO | 0.7557 | 0.003 | 0.734 (0.0091) | 0.0376 (0.0031) | -5149629.32 |
| SA | 0.4329 | 0.0052 | 0.7122 (0.011) | 0.0428 (0.0013) | N/A |
| LIME | 0.3331 | 0.0011 | 0.4214 (0.0803) | 0.0181 (0.0008) | N/A |

Table 2: Simulated Data I & II - Explanation performance compared to ground–truth. For Simulated Data II, we also show in the third column that the log–probabilities of our counterfactuals are higher under the true distribution.

## 4.2 SIMULATED DATA II

The first simulation does not necessarily evaluate feature importance under complex state dynamics as is common in applications. In this simulation, we create a dataset with complex dynamics with known ground truth explanations. The dataset consists of multivariate time series signals with 3 features. A Hidden Markov Model with 2 latent states, with linear transition matrix and multivariate normal emission probabilities is used to simulate observations. The the outcome $y$ is a random variable, which, in state 1, is only affected by feature 1 and in state 2, only affected by feature 2. Also, we add non-stationarity to the time series by modeling the state transition probabilities as a function of time.

The ground truth explanation for output at time $T$ is the observation $x_{i,t}$ where $i$ is the feature that drives the output in the current state and $t$ indicates when feature $i$ became important. In a time series setting, a full explanation for outcome at $t = T$ should include the most important feature variable as well as the time point of importance (here state change).

Figure 4 demonstrates assigned importance for a time series sample. The shaded regions indicate the top 5 important observations ($x_{i,t}$) for each method, the color indicating the corresponding feature $i$. AFO, FO and FFC are able to learn the state dynamics and are able to find the important feature of each state. However, the top importance values in AFO and FO do not correspond to the important time points. Only in FFC, the top important observations are indicative of state changes. Table 2 shows the performance compared to ground-truth explanations for this data.

### 4.2.1 EFFECT OF GENERATOR SPECIFICATION

As mentioned earlier, the quality of explanations rely on the quality of the counterfactuals. The counterfactuals should reflect the underlying latent dynamics for an individual sample. Counterfactuals under the marginal (as used by AFO) need not be likely for a specific sample. The conditional distribution we use, on the other hand, models the underlying characteristic of an individual sample, while the marginal is an average over the population. Counterfactuals unlikely under an individual patient's dynamics can result in inaccurate importance assignments since they can potentially overestimate the change in model outcome significantly. We demonstrate this by evaluating the log probability of the counterfactual under the true generator distribution $p^*(\mathbf{x}_t|\mathbf{X}_{0:t-1})$. Results are summarized in Table 2, Column 3. Since we simulate data using an HMM, we can analytically derive the distribution $p^*(x_{i,t}|\mathbf{X}_{0:t-1})$. Details of the derivation are included in Appendix A.2.1.

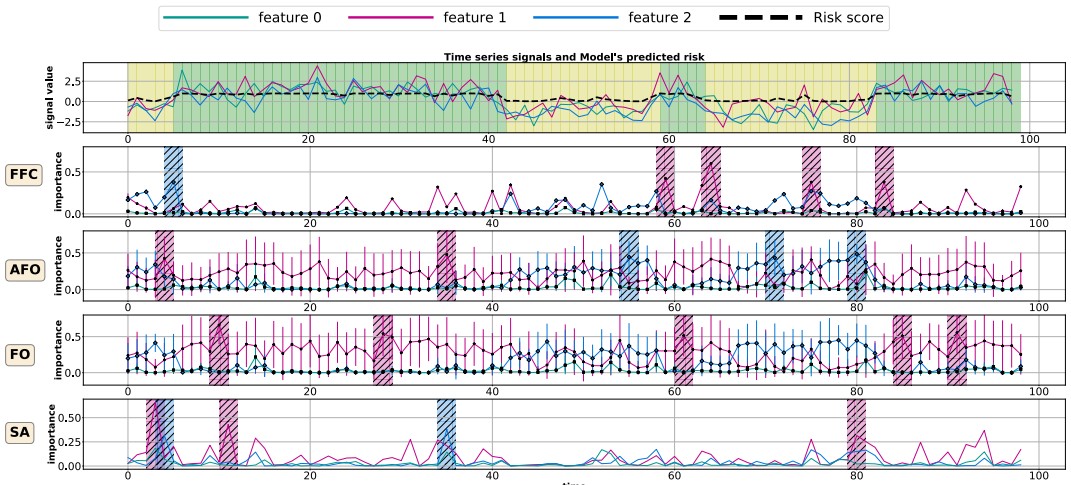

Figure 4: Simulated Data II: The top plot shows the time series signals and the output risk of the black-box model. States are shaded in green and yellow. Plots below show the reported importance of all features over time for baseline methods. Shaded regions in these plots represent the top 5 important observations reported by each method.

## 4.3 MIMIC MORTALITY

Explaining models based on feature importance is critical for clinical settings. Most of clinical data come in form of time series therefore, in is important to find critical time points in patients' trajectories along with important features. We evaluate our method on a benchmark mortality prediction task based on the Intensive Care Unit (ICU) time series data from MIMIC. The MIMIC-III dataset consists of de-identified EHRs for $\sim 40,000$ ICU patients at the Beth Isreal Deaconess Medical Center, Boston, MA. The dataset has time series measurements such as vitals and lab results over patients ICU stay (Johnson et al., 2016). We use an RNN-based mortality predictor model as a black-box for evaluation. The model is trained on $14,712$ adult ICU admissions and reaches a classification AUC of $0.7939(0.007)$, using 8 vital and 20 lab measurements, as well as patient static data. More details on the model and data used are in Appendix A.3.

Following the procedure in Algorithm 1, we train a conditional generators for non-static time series features. We compare results across all four existing methods by visualizing importance scores over time. Figure 5 shows an example trajectory of a patient and the predicted outcome. We plot the importance score over time for top 3 signals, selected by each method. Shaded regions in bottom four rows indicate the most important observations, color representing the feature. As shown in Figure 5, counterfactual based methods mostly agree and pick the same signals as important features. We further evaluate this by looking into accordance scores among methods, indicating the percentage of agreement. This analysis is provided in the Appendix A.3, and the heat map in Figure 10 demonstrates the average score across test samples. However, the methods don't agree on the exact time importance. As we observe in Figure 5 and other patient trajectories, FFC assigns importance to observations at the precise times of signal change. This is exactly as expected from the method. The FFC counterfactual is conditioned on patient history and thus the counterfactual represents an observation assuming a patient had not change state.

### 4.3.1 EVALUATION USING CLINICAL ANNOTATIONS

Since evaluation of explanations can be subjective, we also use intervention information present in patient records to evaluate clinical applicability across baselines. Clinicians intervene with a medication or procedure when there is a notable, adverse change in patient trajectory. Therefore, picking up the most relevant features before an intervention onset is indicative of clinical validity of the method. While we cannot directly associate an intervention with a specific cause (observation), we look at the overall distribution of signals considered important by each of the methods, prior to intervention onset. Figure 6 shows these histograms for a number of interventions. We see consistent

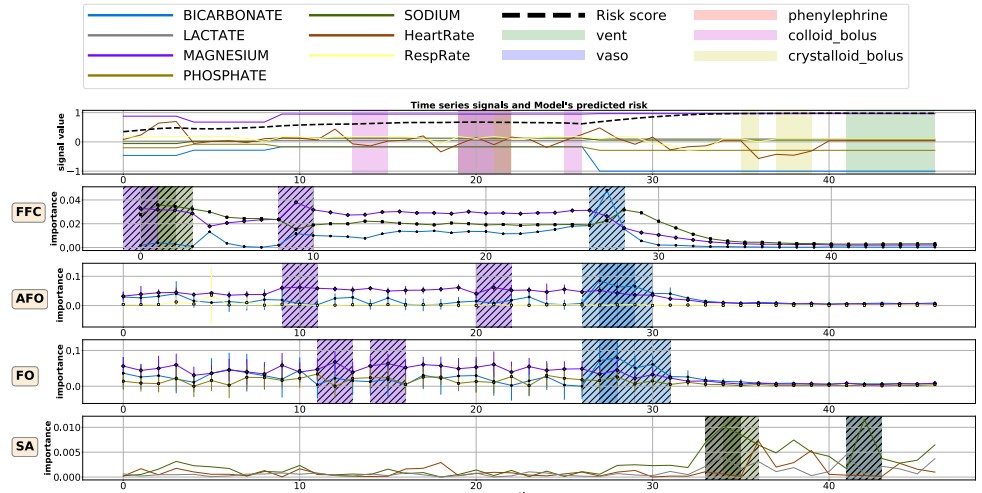

Figure 5: The top row shows the output risk of the prediction model (black dashed line), and normalized time series signals. Shaded regions in the top row represent specific clinical interventions. The four bottom rows correspond to importance scores generated by each of the methods. The error bars indicate the standard deviation of the importance value. Hatched regions in bottom four rows indicate the top most important observations.

assignment of importance across all methods. This means they associate the same influential signals to the same medical intervention.

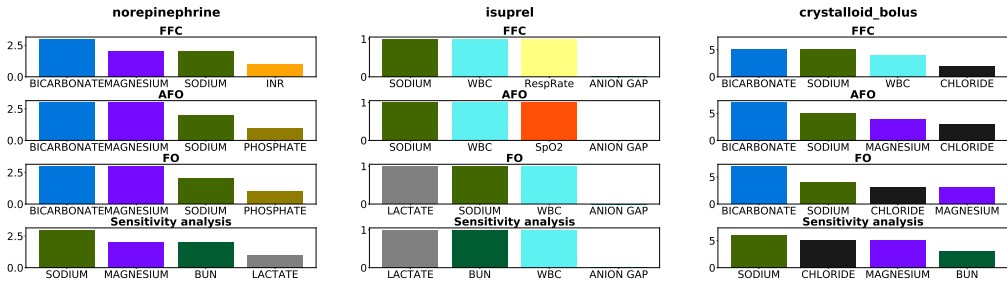

Figure 6: Top four features assigned to be important for each intervention across compared methods.

## 4.4 Greenhouse Gas (GHG) Observing Network Data Set

This experiment evaluates the utility of our method for attributing importance to GHG tracers across different locations in California. The GHG data consists of 15 time series signals from 2921 grid cells. A target time series is a synthetic signal of GHG concentrations. We use an RNN model to estimate GHG concentrations using all tracers. Evaluating which tracers are most useful in reconstructing this synthetic signal can be posed as a feature importance problem for weather modeling over time.

In order to quantitatively evaluate the proposed method on real data, we evaluate how well the method performs at selecting *globally* relevant methods as a proxy. We aggregate the importance of all features over time (and training samples) and retrain the black–box by i) removing top 10 relevant features as indicated by each method ii) using top 3 relevant features only . The performance summary is provided in Table 3 suggesting that among methods that derive instance wise feature importance over time, FFC also generates reasonable global relevance of features. Results for both MIMIC-III and GHG datasets are summarized in Table 3.

| | MIMIC | | GHG | |
|---|---|---|---|---|
| **Method** | AUROC -model w/o top 10 (lower is better) | AUROC -model w/ top 3 only (higher is better) | MSE -model w/o top 10 (higher is better) | MSE -model w/ top 3 only (lower is better) |
| FFC | 0.7807 (0.0025) | **0.7213 (0.0014)** | **4546.460** | 4537.188 |
| AFO | 0.7861 (0.0055) | 0.7208 (0.002) | 4544.163 | **4532.027** |
| FO | 0.0.7861 (0.0026) | 0.72 (0.0021) | 4538.940 | 4537.044 |
| SA | **0.7788 (0.004)** | 0.6939 (0.0011) | 4532.440 | 4552.386 |
| LIME | 0.7798 (0.0031) | 0.6795 (0.0004) | 4536.328 | 4551.874 |

Table 3: Real Data - Global Importance.

| | Data Randomization Test | | Model Randomization Test | |
|---|---|---|---|---|
| Method | Δ AUROC | Δ AUPRC | Δ AUROC | Δ AUPRC |
| FFC | -0.2888 | -0.2129 | -0.2351 | -0.2202 |
| AFO | -0.2060 | -0.0184 | -0.1662 | -0.0174 |
| FO | -0.2070 | -0.0176 | -0.1565 | -0.0172 |
| SA | -0.2252 | -0.0258 | -0.3501 | -0.0253 |

Table 4: Sanity Check Test Results for Simulated Data II. For all measures higher difference is better.

## 4.5 SANITY CHECK TEST FOR EXPLANATIONS

We additionally evaluate the quality of the proposed FFC method using the randomization tests proposed as 'Sanity Checks' in Adebayo et al. (2018). Two randomization tests are designed to test for sensitivity of the explanations to i) the black–box model parameters using a *model parameter randomization test*, and ii) sensitivity to data labels using a using a *data randomization test*. We conduct this evaluation for Simulation Data II.

1. **Data Parameter Randomization Test:** This test evaluates how different explanations are when the black–box model is trained on permuted labels (breaking the correlation between features and output label). If explanations truly rely on the output labels, as suggested in our definition, then the explanation quality should differ significantly when a model trains on permuted labels. We evaluate the drop in the AUROC and AUPRC of the generated explanations compared to the ground truth.

2. **Model Parameter Randomization Test:** This test evaluates how different explanation quality is when the parameters of the model are arbitrarily shuffled. Significant differences in generated explanations suggests the proposed method is sensitive to black-box model parameters. In Adebayo et al. (2018), these tests are conducted for saliency map methods for images by evaluating the distance between saliency maps for different randomizations. The results are included for Simulated Data II, measured with AUROC and AUPRC as ground–truth explanations are available.

The results of both tests are included in Table 4. They indicate the drops in explanation performance for both randomization tests. The performance of the model used for model randomization test drops to 0.52 AUROC as opposed to 0.95 for the original trained model on this simulated task (Simulation Data II). For data randomization, performance of the model drops to 0.62 from 0.95 in terms of AUROC. AUROCs and AUPRCs for FFC drop the most, suggesting the FFC explanation method is sensitive to perturbations in output labels (as tested by the data randomization test) and to randomization in model parameters. Significant deterioration compared to explanation performance in Table 2 (for Simulation Data II) indicates that the proposed method passes the sanity checks.

## 5 DISCUSSION

We propose a new definition for obtaining sample-based feature importance for high-dimensional time series data. We evaluate the importance of each feature at every time point, to locate highly important observations. We define important observations as those that cause the biggest change in model output had they been *different* from the actual observation. This counterfactual observation is generated by modeling the conditional distribution of the underlying data dynamics. We propose a generative model to sample such counterfactuals. We evaluate and compare the proposed definition

and algorithm to several existing approaches. We show that our method is better at localizing important observations over time. This is one of the first methods that provides individual feature importance over time. Future extension to this work will include analysis on real datasets annotated with feature importance explanations. The method will also be extended to evaluate change in risk based on most relevant subsets of observations.

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

# A  APPENDIX

## A.1  SIMULATED DATA I

To simulate these data, we generate $d = 3$ (independent) sequences as a standard non–linear auto-regressive moving average (NARMA) time series. Note also that we add linear trends to features 1 and 2 of the form: $x(t+1) = 0.5x(t) + 0.5x(t) \sum_{i=0}^{l-1} x(t-l) + 1.5u(t-(l-1))u(t) + 0.5 + \alpha_d t$ for $t \in [80]$, $\alpha > 0$ (0.065 for feature 2 and 0.003 for feature 1), and where the order $l = 2$, $u \sim \text{Normal}(0, 0.03)$. We additionally add linear trends to features. We add spikes to each sample (uniformly at random over time) and for every feature $d$ following the procedure below:

$$y_d \sim \text{Bernoulli}(0.5);$$

$$\eta_d = \begin{cases} \text{Poisson}(\lambda = 2) & \text{if } \mathbf{1}(y_d == 1) \\ 0 & \text{otherwise} \end{cases} \quad (2)$$

$$\mathbf{g}_d \sim \text{Sample}([T], \eta_d); \; x_{d,t} = x_{d,t} + \kappa \, \forall t \in \mathbf{g}_d$$

where $\kappa > 0$ indicates the additive spike. The label $y_t = 1 \, \forall t > t_1$, where $t_1 = \min g_d$, i.e. the label changes to 1 when a spike is encountered in the first feature and is 0 otherwise.

We sample our time series using the python TimeSynth[1] package. Number of samples generated: 10000 (80%,20% split).

### A.1.1  ADDITIONAL SAMPLES

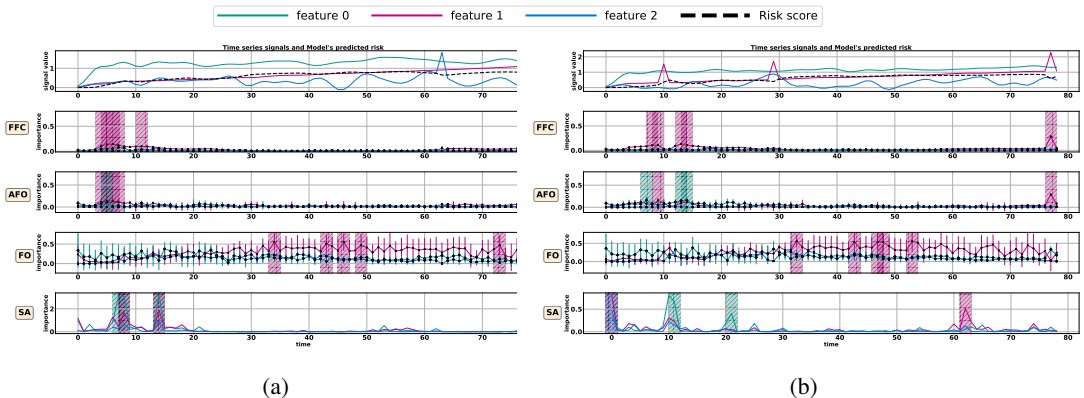

(a)                                          (b)

Figure 7: Two samples showing performance on simulated data I. Top row are the original sampled signals (one per subfigure).

## A.2  SIMULATED DATA II

This simulated data is a two state HMM (2 states) with initial state $\pi = [0.5, 0.5]$. Transition probability $T$ being:

$$T = \begin{bmatrix} 0.1 & 0.9 \\ 0.1 & 0.9 \end{bmatrix}$$

The emission probability in each state is a multivariate Gaussian: $\mathcal{N}(\mu_1, \Sigma_1)$ and $\mathcal{N}(\mu_1, \Sigma_1)$ where $\mu_1 = [1.2, 1.5, 0.8]$ and $\mu_2 = [-1.2, -0.8, -1.5]$. Marginal variance for all features in each state is 0.8 with only features 1 and 2 being correlated ($\Sigma_{12} = \Sigma_{21} = 0.01$) in state 1 and only 0 and 2 om state 2 ($\Sigma_{02} = \Sigma_{20} = 0.01$). In state 1, the label $y$ only depends on feature 1 and in state 2, label depends only on feature 2.

---

[1] https://github.com/TimeSynth/TimeSynth

The output $y_t$ at every step is assigned using the $logit$ in 3. Depending on the hidden state at time $t$, only one of the features contribute to the output and is deemed influential to the output.

$$p_t = \begin{cases} \frac{1}{1+e^{-x_{1,t}}} & s_t = 0 \\ \frac{1}{1+e^{-x_{2,t}}} & s_t = 1 \end{cases} \qquad (3)$$
$$y_t \sim Bernoulli(p_t)$$

### A.2.1 DERIVATION OF $p^*(x_{i,t}|\mathbf{X}_{0:t-1})$

The true conditional distribution can be derived using the forward algorithm (Bishop, 2006) as follows:

$$p^*(x_{i,t}|\mathbf{X}_{0:t-1}) = \sum_{s_t \in \{0,1\}} p(x_{i,t}|s_t)p(s_t|\mathbf{X}_{0:t-1}) \qquad (4)$$

where,

$$p(s_t|\mathbf{X}_{0:t-1}) = \sum_{s_{t-1} \in \{0,1\}} p(s_t|s_{t-1})p(s_{t-1}|\mathbf{X}_{0:t-1}) \qquad (5)$$

where $p(s_{t-1}|\mathbf{X}_{0:t-1})$ is estimated using the forward algorithm.

### JOINT CONDITIONAL GENERATIVE MODEL

Our generator $\mathcal{G}_i$ is trained using an RNN (GRU). We model the latent state $\mathbf{z}_t$ with a multivariate Gaussian with diagonal covariance and observations with a multivariate Gaussian with full covariance. Parameter setting of the generator for each of the experiments are shown in tables below.

| Setting | value |
|---|---|
| RNN cell | GRU |
| Loss | MSE |
| Optimizer | Adam |

Table 5: General generator Setting

Software used: Python 3.7.3 , Pytorch 1.0.1.post2

GPU Info: Quadro 400

CPU Info: Intel(R) Xeon(R) CPU E5-1620 v4 @ 3.50GHz

The counterfactual for observation $i$ at time $t$ can now be sampled by marginalizing over other features at time $t$. i.e, $x_{i,t} \sim \sum_{\mathbf{x}_{-i}} p(\hat{\mathbf{x}}|\mathbf{X}_{0:t-1})$.

### A.3 MIMIC-III MORTALITY EXPERIMENT

**Feature selection and data processing:** For this experiment, we select adult ICU admission data from the MIMIC dataset. We use static patients' static, vital measurements and lab result for the analysis. The task is to predict 48 hour mortality based on 48 hours of clinical data, therefor we remove samples with less than 48 hours of data. Table 6 presents a full list of clinical measurements used in this experiment.

The predictor model takes in new measurements every hour, and updates the mortality risk. We quantize the time series data to hour blocks by averaging existing measurements within each hour block. We use 2 approches for imputing missing values: 1) Mean imputatiopn for vital signals using the sklearn SimpleImputer [2], 2) forward imputation for lab results, where we keep the value of the last

---

[2]https://scikit-learn.org/stable/modules/generated/sklearn.impute.SimpleImputer.html

lab measurement until a new value is evaluated. We also removed patients who had all 48 quantized measurements missing for a specific feature. Overall, 22,988 ICU admissions were extracted and training process was on a 65%,15%,20% train, validation, test set respectively.

| Data class | Name |
|---|---|
| Static measurements | Age, Gender, Ethnicity, first time admitted to the ICU? |
| Lab measurements | LACTATE, MAGNESIUM, PHOSPHATE, PLATELET, POTASSIUM, PTT, INR, PT, SODIUM, BUN, WBC |
| Vital measurements | HeartRate, DiasBP, SysBP, MeanBP, RespRate, SpO2, Glucose, Temp |

Table 6: List of clinical features for the risk predictor model

**Parameter Settings for mortality risk predictor model:** The risk predictor model is a recurrent network with GRU cells. All features are scaled to 0 mean, unit variance and the target is a probability score ranging $[0, 1]$. The model achieves $0.7939(0.007)$ AUC on test set classification task. Detailed specification of the model are presented in Table 7.

| Setting | value |
|---|---|
| epochs | 80 |
| Model | GRU |
| batch size | 100 |
| Encoding size ($m$) | 150 |
| Loss | MSE |
| Regressor Activation | Sigmoid |
| Batch Normalization | True |
| Dropout | True |
| Gradient Algorithm | Adam (learning rate $= 0.001$, $\beta_1 = 0.9$, $\beta_2 = 0.999$, weight decay $= 0$) |

Table 7: Mortality risk predictor model features

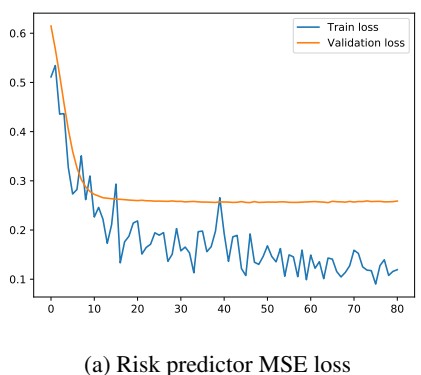

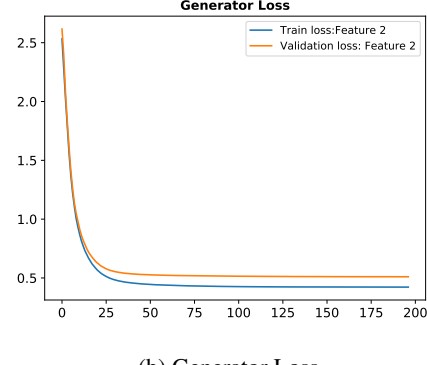

(a) Risk predictor MSE loss

(b) Generator Loss

**Parameter Settings for conditional Generator:** The recurrent network with specifications show in 8 learns a hidden latent vector $h_t$ representing the history. $h_t$ is then concatenated with $x_{-i,t}$ and fed into a non-linear 1-layer MLP to model the conditional distribution $p(x_i, t|\mathbf{X}_{0:t-1})$.

Additional importance plots are provided in Figure 9.

| Setting | value |
|---|---|
| epochs | 150 |
| RNN cell | GRU |
| batch normalization | True |
| batch size | 100 |
| RNN encoding size ($m$) | 80 |
| Regressor encoding size ($m$) | 300 |
| Loss | MSE |
| Gradient Algorithm | Adam (learning rate $= 0.0001$, $\beta_1 = 0.9$, $\beta_2 = 0.999$, weight decay $= 0$) |

Table 8: Training Settings for Feature Generators for MIMIC-III Data

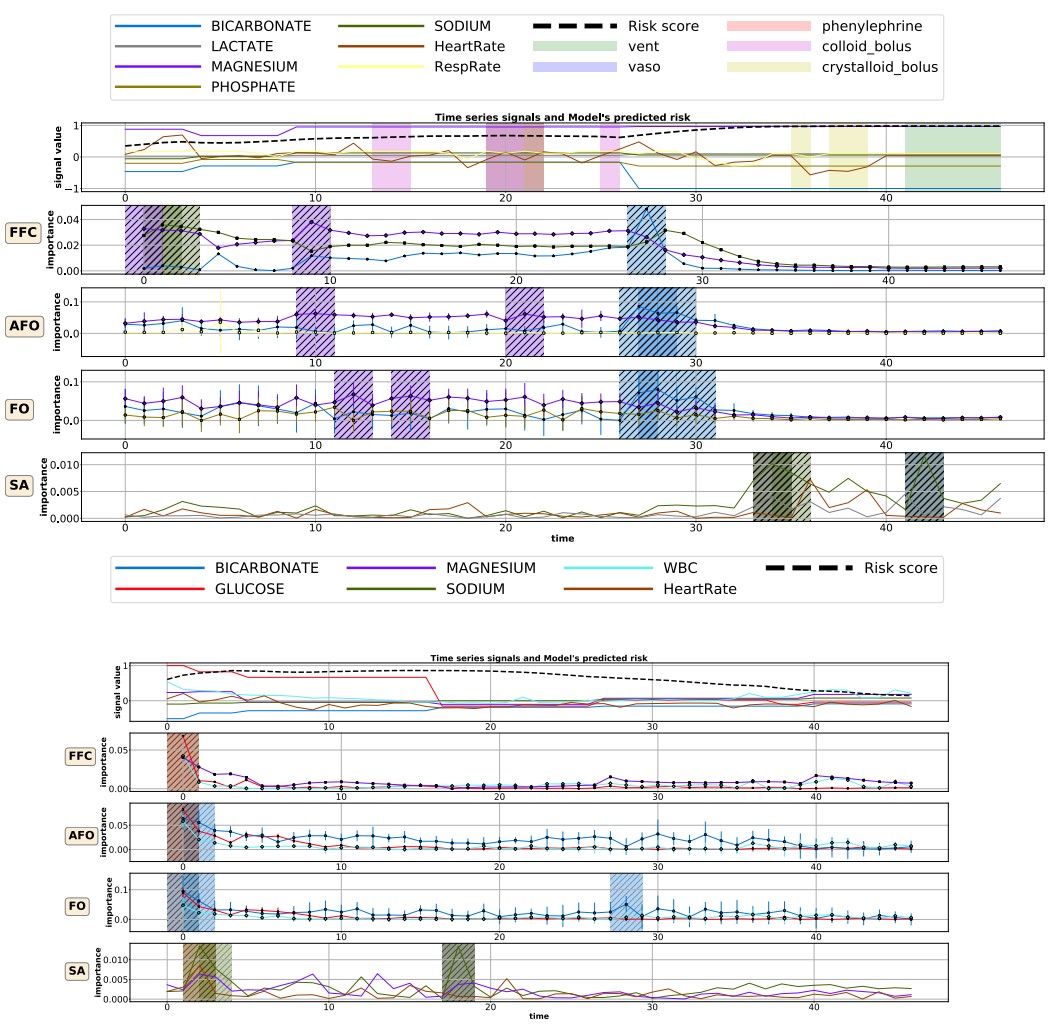

Figure 9: Additional patient trajectories and feature importance assignment with all baseline methods

**Accordance testing:** For this test we look into how much different baselines agree on important feature assignment. As we observed from the experiments, counterfactual methods mostly agree on the most important features for individual samples. We define accordance score between 2 methods as the percentage of top $n$ signals both identified as important. A score of 80 means on average over the test data, 80 of the assignments were similar. This is depicted in Figure 10.

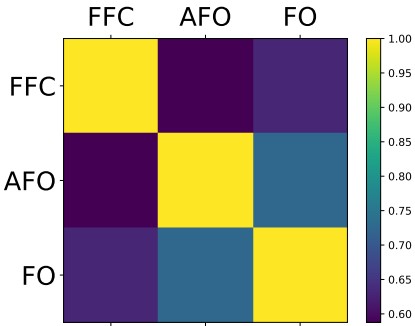

Figure 10: Heat map showing the accordance score between pairs of methods for the most important 6 signals out of 31 clinical features

### A.4 RUN-TIME ANALYSIS

In this section we compare the run-time across multiple baselines. Table 9 shows inference runtime for all the baseline methods on a machine with Quadro 400 GPU and Intel(R) Xeon(R) CPU E5-1620 v4 @ 3.50GHz CPU. The runtime for the counterfactual approaches (FFC, FO and AFO) depends only on the length of the time series. It is also the case for FFC since the conditional generator models the joint distribution of all features. This is an advantage, over approaches like LIME, the runtime depends both on the length of the signal as well as the number of features.

Overall, FFC performs reasonably compared to ad-hoc counterfactual approaches, since inference on the RNN conditional generator is efficient. This is one of the reasons that the RNN generator model is used to approximate the conditional distribution.

| Method | Simulation data $t = 100, d = 3$ | MIMIC data $t = 48, d = 27$ |
|---|---|---|
| FFC | 0.99 | 0.36 |
| AFO | 1.64 | 0.62 |
| FO | 2.09 | 0.84 |
| LIME | 2.23 | 8.72 |
| Sensitivity Analysis | 0.212 | 0.055 |

Table 9: Run-time results for simulated data and MIMIC experiment.

### A.5 GHG NETWORK DATA

**Parameter Settings for Generator:**   The settings are provided in Table 10.

| Setting | value |
|---|---|
| epochs | 200 |
| RNN cell | GRU |
| batch size | 100 |
| Encoding size ($m$) | 100 |
| Loss | MSE |
| Gradient Algorithm | Adam (learning rate $= 0.0001$, $\beta_1 = 0.9$, $\beta_2 = 0.999$, weight decay $= 0$) |

Table 10: Training Settings for Feature Generators for GHG Data

**Parameter Settings for Black-Box:**   This black box regresses $d = 15$ tracer time signals to the target synthetic GHG time series for $t = 327$ time points. This model is trained using a 65%,15%,20% train, validation, test set respectively. All features are scaled to 0 mean unit variance and the target

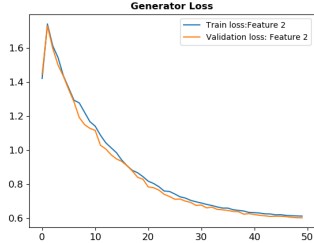 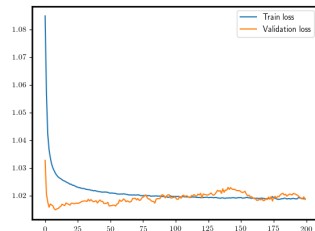

Figure 11: Left: Generator Loss for ghg data. Right: (Scaled) Regresser MSE loss

is scaled time series is scaled in the range $[-1, 1]$. The regressor is an RNN model with the parameter settings given in Table 11. Figure 11 (a) shows the generator loss for all trained conditional

| Setting | value |
|---|---|
| epochs | 200 |
| Model | RNN |
| batch size | 100 |
| Encoding size ($m$) | 100 |
| Loss | MSE |
| Regressor Activation | Linear |
| Gradient Algorithm | Adam (learning rate $= 0.001$, $\beta_1 = 0.9$, $\beta_2 = 0.999$, weight decay $= 0$) |

Table 11: Training Settings for Regressor for GHG Data

(counterfactual) generators, while Figure 11 shows the training loss of black-box that was used to present feature important results in Section 4.4.

