# OpenReview forum: "Explaining Time Series by Counterfactuals"
_ICLR.cc/2020/Conference — Reject_

### Official Review · AnonReviewer1 · 2019-10-21
**Official Blind Review #1**

**Rating:** 3

**Review:**

This paper proposes an extension of feature occlusion (FO) [Suresh et al., 2017] in which they sample from a pre-trained generative model for replacing the observed variables. Technically, it is in the category of saliency maps, only with possibly larger perturbations defined by the generative model. Thus, it most likely should inherit the same properties as the saliency maps.

The authors experiment on both synthetic and real-world datasets. They use log-probabilities of the generated samples as a metric for the quality of the counterfactuals; however, because not all baselines are based on counterfactuals, this approach has limited usefulness. Also, it seems that the log-probabilities are too small, indicating that most likely the authors have reported the sum log-probabilities instead of average probabilities.

Given the similarity to the saliency maps, the authors should have tested the proposed method in the sanity checks in [1]. Also, the authors should have examined the robustness of the proposed explanations given the adversarial vulnerability phenomenon.

Despite sampling from a generative model, because of the univariate nature of the counterfactuals used in this paper, the process might create invalid data points. For example, increased blood sugar is usually correlated with increased blood pressure. However, this method does not account for the correlation among the features.

Finally, this method should be very slow to run. The authors should have compared the run-time speed of the algorithms in the experiment section, too.

[1] Adebayo, J., Gilmer, J., Muelly, M., Goodfellow, I., Hardt, M., & Kim, B. (2018). Sanity checks for saliency maps. In NeurIPS.

**Experience Assessment:**

I have published one or two papers in this area.

**Review Assessment: Checking Correctness Of Derivations And Theory:**

I carefully checked the derivations and theory.

**Review Assessment: Checking Correctness Of Experiments:**

I assessed the sensibility of the experiments.

**Review Assessment: Thoroughness In Paper Reading:**

I read the paper at least twice and used my best judgement in assessing the paper.

---

> ### Author Response · Authors · 2019-11-13
> **Thank you for your feedback**
>
> 1. Saliency test:
> We agree that adding ‘sanity checks for saliency maps’ is a great idea, and have done so. In addition to presenting the results here, we have also added a corresponding section in the paper. Here, we report the results of 3 randomization tests on our simulation data II to evaluate the robustness of our approach. The results further support our claim that our explanations are expectedly sensitive to model parameters and the relationship between the input and labels. Thus our FFC method passes the model and data randomization tests.
>
> -- Data randomization test:
> For this experiment, we train the model on data with shuffled labels (predictor model AUC is 0.62). The table reports the drop in explanation performance for all baselines. We see that FFC has the greatest drop compared to the original model. An example of explanations generated for the trained and random model is shown in the figure (“randomized_data.pdf” [1])
> -- Model randomization test:
> We randomly shuffle the parameters of the prediction model and inspect the effect on the explanations. The shuffled predictor model has an AUC of 0.52. Figure (“randomized_param.pdf” [1]) shows an example of importance assignment results for the randomized model as well as the trained models. We can see that the explanation results are different for the 2 different models in this example. The overall performance drop for the randomized model is also reported in the table below
>
>                           Data Randomization Test                         Model Randomization Test
> Method   | $\Delta$ AUROC  |    $\Delta$ AUPRC  | $\Delta$ AUROC  | $\Delta$ AUPRC
> ____________________________________________________________________________|__________________
>   FFC         | -0.2888                 | -0.2129                     | -0.2351                    | -0.2202
>  AFO         | -0.2060                 | -0.0184                     | -0.1662                    | -0.0174
>   FO          | -0.2070                 | -0.0176                     | -0.1565                   | -0.0172
>   SA           | -0.2252                 | -0.0258                     | -0.3501                    | -0.0253
>
> 2. Runtime analysis:
> Thank you for bringing this up. In the table below we report inference runtime (in seconds) for all the baseline methods on a machine with Quadro 400 GPU and Intel(R) Xeon(R) CPU E5-1620 v4 @ 3.50GHz CPU. The runtime for the counterfactual approaches (FFC, FO, and AFO) is only dependant on the length of time series. This is clear for AFO and FO, but it is also the case for FFC since the conditional generator models the joint distribution of all features. This property is an advantage since, for approaches like LIME, the runtime depends both on the length of the signal as well as the number of features. Overall, FFC performs reasonably compared to ad-hoc counterfactual approaches, since inference on the RNN based conditional generator is efficient. This is one of the reasons that the RNN generator model is used to approximate the conditional distribution. We have also added this analysis to the supplementary material of the paper.
>
>  Method | Simulation data(t=100, d=3) | MIMIC data(t=48, d=27)
>    FFC      |                  0.99                          |                  0.36
>    AFO     |                 1.64                           |                  0.62
>    FO        |                 2.09                           |                 0.84
>    LIME    |                 2.23                           |                 8.72
>    SA        |                0.212                          |                 0.055
>
> 3.Log-probabilities:
> Log-probabilities represent the likelihood of a sample under the original data distribution. We have used them only as a measure to evaluate the quality of the generators. These values are not reported to compare explanation qualities across methods and we will ensure we clarify this in the text.
>
> 4.Adversarial attacks on explanations:
> We would appreciate it if you could clarify your concern in this regard. In our understanding, the robustness of explanations would depend on the robustness of the prediction model on adversarial attacks. However, investigating the explanation for non-robust models under adversarial attacks is an interesting extended analysis of our method. We will actively consider this as future work since it can be a valuable standalone contribution.
>
> 5.Univariate nature of the counterfactuals:
> As stated in the future work section of the document, our next steps are to extend this method to find subsets of features with the highest importance. This involves sampling multivariate counterfactuals like you suggest based on feature correlations. Enumerating all possible subsets over features to estimate importance is inefficient. We are actively considering follow up work that will allow doing this efficiently.
>
> [1] https://www.dropbox.com/sh/tpfci7w1tqc3id5/AACwtkwqBUuGnZU8xqfldjyNa?dl=0

---

### Official Review · AnonReviewer2 · 2019-10-24
**Official Blind Review #2**

**Rating:** 6

**Review:**

--- Overall ---

This paper proposes a method for evaluating the influence of individual observations on the output of a time series prediction model by replacing each (discrete time) observation with its conditional expectation given the other observations. They evaluate this method qualitatively on synthetic, healthcare, and climate datasets. I reviewed this paper for NeurIPS and was happy to see that the authors have made substantial improvements to the presentation and evaluation of the method. With that said, I think that the methodological contribution is incremental (sampling from a conditional rather than marginal distribution), there is at least one major correctness issue that needs to be addressed, and the analysis of the experiments fails explain why the models perform differently.

--- Major comments ---

1. The Montecarlo approximation in Algorithm 1 does not approximate Imp(i,t). Specifically, because the averaging is done before the absolute value, Algorithm 1 approximates |F(X_{0:t}) - E[F(X_{0:-t},x_{-i,t},\hat{x}_{i,t}]|. This is also a valid measure of feature importance and it is not clear from the paper why we should prefer one over the other.

2. I think the paper needs to do a better job explaining why sampling from the conditional leads to better explanations than sampling from the marginal. The second paragraph makes an argument based on variance, but it is not clear that low variance translates to better explanations. In particular, using mean imputation has very low variance, but I would expect it to give poor explanations. I recommend using a toy example to make this point. For example, in a healthcare context, doctors are reacting to changes relative to a particular patient's baseline. A conditional model can capture this baseline but a marginal model cannot.

3. In general, I thought that the experiments were well done, but stops short of explaining *why* the methods perform differently. Put differently, I think it is really important to clearly explain why certain methods fail while others succeed. For example, the authors demonstrate the sensitivity analysis fails on the synthetic data, but never explain why. I am looking for a statement of the form: "Sensitivity analysis fails on this data because... FFC solves this weakness by doing... which is reflected in the experimental results."

4. In 4.2.1, it is very unsurprising to me that a model that samples from an approximation of the conditional has higher likelihood under the conditional than samples from the marginal, but why should we expect this to lead to better explanations?

5. I thought the idea of looking at feature importance just before clinician intervention was a very clever evaluation, but I wanted the qualitative evaluation to go one step further. That is, does bicarbonate being the most important feature just before administration norepinephrine and fluids make clinical sense? Is this picking up on a specific condition and if so what condition? A clinician could tell you what they are typically reacting to when they administer fluids or vasopressors and you can compare what they say to what the model says. I was surprised to see the top features all being lab measurements as opposed to vital signs. In particular, in a vacuum, I would expect systolic blood pressure to be the most important feature in both of these cases. Is it possible that the frequency of measurements affects which features are selected as important?

6. I thought GHG experiment was *much* better and clearer in this version of the paper. Well done.

7. I recommend moving the notation from the appendix to the main paper. I don't think a reader should have to reference another document to follow notation.

--- Minor comments ---

1. Pg. 3 "The magnitude of our...": I call the authors' definition of feature importance absolute not relative. I would expect a "relative importance" to be a ratio of some sort (e.g. relative risk).

2. Pg. 5: Figure 8 --> Figure 3

**Experience Assessment:**

I have read many papers in this area.

**Review Assessment: Checking Correctness Of Derivations And Theory:**

I carefully checked the derivations and theory.

**Review Assessment: Checking Correctness Of Experiments:**

I carefully checked the experiments.

**Review Assessment: Thoroughness In Paper Reading:**

I read the paper thoroughly.

---

> ### Author Response · Authors · 2019-11-13
> **Thank you for your feedback**
>
> We would like to thank the reviewer for taking the time to provide thoughtful and constructive feedback on our paper. We addressed all your comments and believe it made our paper better in the process.
>
> 1. Definition:
> Thank you very much for spotting the discrepancy in our two definitions of importance. There indeed was a typo in the algorithm.  The difference primarily occurs when the counterfactual sample is very close to the actual observation. In this situation, averaging before evaluating the absolute value may underestimate the amount of possible risk change we have observed. Therefore we think that definition 1 should be preferred.  We have fixed the algorithm box and code to match and updated all results in the paper (along with making the simulations more realistic based on other reviews), and this did not change the relative performance of our method.
>
>
> 2. Choice of conditioning:
> We agree and updated the paper to clarify the benefit of using the conditional distribution versus the marginal.
> The conditional distribution we use models the underlying characteristic of an individual sample, while the marginal is an average over the population. Counterfactuals under the marginal distributions may not necessarily be likely or realistic for a specific sample, as reflected in log-probabilities. Unrealistic counterfactuals can result in inaccurate importance assignments since they can potentially overestimate the change in model outcome significantly, but only because they are unlikely under the individual sample’s distribution.
>
> 3. Explanation of baseline failures:
> Sensitivity analysis characterizes the approximate change in the risk due to relatively infinitesimally small perturbations to the observations. Such estimates are unreliable for levels of perturbations observed when a patient changes state or deteriorates significantly. We see this unreliability in the Simulation Experiment II (Figure 2) where the underlying generative model has latent states (HMM). The meaningful time and feature importance correspond to that of state transitions while sensitivity analysis highlights within state variations. FFC resolves this issue by evaluating risk changes on perturbations that are clearly indicative of past patient state and could be significantly large if the underlying state has changed. By virtue of the design of RNN based methods, sensitivity analysis is also more likely to highlight more recent observations as important, as we see in MIMIC-III experiments. By observing local estimates in risk changes, FFC and other methods avoid this issue. Finally, we do not believe it is entirely fair to LIME to compare it to methods like FFC, and AFO, that are specially designed for time series data. LIME locally approximates the model around the current sample to determine the importance and thus uses much less dynamic information than methods designed specifically for time series, leading it to perform much worse than all other methods.
>
>
> 4. Log-likelihood and Better explanations:
> We have only used the log-probabilities as a measure to evaluate the quality of generated counterfactuals. These values are not reported to compare explanation qualities across methods and we will ensure that this is clarified in the text. The relationship between choice of modeled distribution and explanation is elaborated above in "Choice of conditioning"
>
> 5. Clinical annotations:
> We will obtain additional clinical annotations based on your suggestions, and as it was also mentioned in our future work section. The difficulty with this task is that clinicians often don’t associate direct relationships between observations and interventions. However, we are hoping to find some accurate and generalizable annotations by interviewing a larger group of clinicians, and aggregating results across.
>
> We agree that the frequency of measurements is related to which features are deemed more important. However, this is due to the way the model learns to predict a risk change. We are therefore limited by the data as well as the model in terms of the quality of the explanation. All explanation baselines we compare to will suffer from this limitation as well. Thank you for bringing this up as we consider it an important followup evaluation to characterize how the frequency of observations affect importance estimates.

---

### Official Review · AnonReviewer3 · 2019-10-25
**Official Blind Review #3**

**Rating:** 3

**Review:**

This paper presents a new method computing the importance of features in time series, called Feed Forward Counterfactual (FFC).
In previous work, the explainability problem in time series was tackled with feature occlusion (FO), sensitivity analyses (SA) methods. However, previous counterfactual based methods do not carefully consider appropriate conditional distribution and generate out-of-distribution counterfactuals.
The proposed FFC method addresses this issue by leveraging a generative model which learns the underlying dynamics and generates a realistic counterfactual given the past observations. FFC is evaluated on simulated and real datasets and shows that it is better at localizing important observations over time compared to the other baselines.
In summary, this paper introduces a way of defining the feature importance at every time point. The main idea of this paper follows in line with [Chang et al. 2019] which address the problem of out-of-distribution counterfactuals. Although the experiment shows successes of the proposed method on several datasets, the major weakness of this paper is the lack of technical novelty and detailed analysis of the proposed method. For example,
If the time series is non-stationary, this could incur a different amount of the change in the model output and proposed time importance might not work. How about this?
Did the authors consider trying out with varying size of training data or generator model?
Minor
On page 2, p(\mathbf{x_{t,i}|\mathbf{X_{0:{t-1}}}}) -> p(x_{i,t}|\mathbf{X_{0:{t-1}}}})
On page 6, affected by feature and -> affected by feature 1 and


**Experience Assessment:**

I have published in this field for several years.

**Review Assessment: Checking Correctness Of Derivations And Theory:**

I assessed the sensibility of the derivations and theory.

**Review Assessment: Checking Correctness Of Experiments:**

I assessed the sensibility of the experiments.

**Review Assessment: Thoroughness In Paper Reading:**

I read the paper at least twice and used my best judgement in assessing the paper.

---

> ### Author Response · Authors · 2019-11-13
> **Thank you for your feedback**
>
> We address your concerns in detail below.
>
> 1.Technical novelty:
> We agree our method has little technical novelty but fortunately that was not the goal of this contribution. We purposely used standard models and approaches, because our main contributions are conceptual: We introduce a new approach to explaining model decisions. Our approach generalizes standard saliency map methods, which rely only on gradients. Gradient-based perturbations can be viewed as infinitesimally-different counterfactuals. We approximately integrate over the entire space of counterfactuals to find the data that would, in expectation, most change the decision if it were observed. This definition of counterfactual based explanations is more suitable for a time series domain, as it allows to characterize underlying dynamics in the signal. To the best of our knowledge, this is a substantial contribution to an overlooked problem in modeling time series data and has the potential of being used in a lot of applications, including but not limited to healthcare.
>
> 2. Detailed Analysis:
> We agree that more analysis would shed light on the model. To improve this aspect of the paper, we’ve added these extra analyses in the paper as described below.
> -- Non-stationary time series:
> Your question about nonstationarity is a good one.  Like all model explanation methods, our method’s explanations will depend on whether the model being explained successfully models non-stationarity. Following this thread, we updated our simulation experiment to include non-stationarity by making the transition probability in the HMM a function of time. The table below reports performance results for this data, and appear in the updated draft. We would like to clarify that since the model output is a probability, the scale of the output doesn’t change over time with non-stationarity and this will not be an issue here.
>
> Method  |        AUROC        |         AUPRC
> __________|___________________|_____________________
> FFC        |   0.954 +/- 0.005  |   0.259 +/- 0.035
> AFO      |   0.724 +/- 0.012   |   0.0374 +/- 0.002
> FO        |   0.734 +/- 0.009   |   0.0376 +/- 0.003
> Sens     |   0.712 +/- 0.011   |   0.0428 +/- 0.001
> LIME     |   0.4214 +/- 0.080|   0.0181 +/- 0.0008
>
> Results demonstrate our method works on models trained on non-stationary data. All other baselines deteriorate substantially in this regime compared to their performance on stationary data.
>
> -- Explanation quality as a function of generator quality:
> We agree that evaluating the choice of generators is an interesting question. Different generators can be obtained by varying the data size for training the generator or changing the overall model structure. In our experiments, varying training size does not impact the generator performance significantly. This is mainly because of the time-series nature of our signals. A few samples are enough for the generator to model the dynamics. As shown in Figure (“AUROC_percent.pdf” [1]), we can see that the change in explanation performance is also negligible.
> In terms of different generator models, the AFO method we introduce is another class of generators that samples counterfactuals from a marginal distribution. We also added a generator that only carries forward previous observations based on your feedback. As shown in the table below, using a simpler generator decreases performance. We will investigate other generators to further quantify the quality of explanations as a function of generator quality.
>
> Method                      |  AUROC  |  AUPRC
> ____________________________________________
> FFC                              |   0.954    |   0.259
> FFC-Carry Forward   |   0.8692  |   0.1215
> AFO                             |   0.724    |   0.0374
> FO                                |   0.734    |   0.0376
>
> -- Sanity check
> We have also added another section to our evaluation, called sanity checks (Sec 4.5) as recommended by Reviewer #1. The goal is to evaluate the robustness of explanations in accordance with tests proposed by Adebayo et al, Sanity Checks for Saliency Maps, NeurIPS, 2018. The results further support our claim that our explanations are expectedly sensitive to model parameters and the relationship between the input and labels.
>
> [1] https://www.dropbox.com/sh/tpfci7w1tqc3id5/AACwtkwqBUuGnZU8xqfldjyNa?dl=0

---

### Public Comment · ~Dani_Kiyasseh1 · 2019-10-25
**Simple Yet Interesting Approach and Results - Several Questions**

Implementation Details -

1) Could the authors provide some more details about the RNN conditional generator. This is how I understood it: the outputs of the RNN are a mean vector and covariance matrix which are used to model a multivariate Gaussian. A vector, z_t, is sampled from this Gaussian and concatenated to the input vector at time t (except for feature of interest) to eventually generate scalar counterfactual observation. Please correct my understanding if this is inaccurate.

Definition of Importance -

2) Based on your definition of importance, it appears that it might be sensitive to models that lack robustness to input perturbations. Is there a way to show that this metric leads to consistent feature importance results regardless of the model sensitivity?

MIMIC Results -

3) The word 'importance' is being used loosely. In Figure 5, for instance, it is interesting that the FFC and AFO approaches identify clinically valid important features with regards to the subsequent intervention. The task, however, was mortality prediction. Therefore, are these features important in the context of the overall mortality task or to the intermediate interventions? Perhaps the authors can identify locally and globally-important features. (By global, I mean pertaining to the high-level task of mortality prediction).

4) Would the relative importance of the variables remain the same when the model is trained with fewer clinical parameters? Consistency in this context would be useful in the event certain clinical environments do not have access to/cannot collect all the included variables.

5) In Figure 5, FFC, AFO, and FO all appear to place close-to-zero importance on the features around the onset of a risk-score of 1. Intuitively, this makes sense and if not enforced in any way into the model, is a reassuring outcome of your method.

6) Could the authors shed light on an example on the MIMIC dataset where their approach produces nonsensical results? This would provide future researchers with insight on how to improve upon your approach.

Thank you,

---

> ### Author Response · Authors · 2019-10-28
> **Inline response**
>
> We thank you for your comment. Please see our responses below.
>
> 1) Could the authors provide some more details about the RNN conditional generator.
>
> The RNN conditional generator only models the joint distribution of all the features at every time step using a multivariate gaussian distribution. The trained generator is eventually used to generate the counterfactual vector $\hat{\mathbf{x}}_{t}$. To derive importance for feature $i$ at time $t$, we concatenate this marginal $\hat{x}_{i,t}$ with actual observations $\mathbf{x}_{-i,t}$.
>
> 2) Based on your definition of importance, it appears that it might be sensitive to models that lack robustness to input perturbations.
>
> Our method explains model behavior and as mentioned in the draft is definitely dependent on model performance. If a model lacks robustness, the explanations would be highlight feature importances the model has picked up on. This is not a limitation of the explanation method, but of the model itself.
>
> 3) The word 'importance' is being used loosely. In Figure 5, for instance, it is interesting that the FFC and AFO approaches identify clinically valid important features with regards to the subsequent intervention. The task, however, was mortality prediction. Therefore, are these features important in the context of the overall mortality task or to the intermediate interventions? Perhaps the authors can identify locally and globally-important features. (By global, I mean pertaining to the high-level task of mortality prediction).
>
> The definition of importance is only associated with the mortality task. Interventions as we know generally help stabilize a patient's condition after a deterioration. We use information about interventions to validate our explanations generated for the predictive task.
>
> 4) Would the relative importance of the variables remain the same when the model is trained with fewer clinical parameters? Consistency in this context would be useful in the event certain clinical environments do not have access to/cannot collect all the included variables.
>
> The process of generating explanations using the conditional generator is completely decoupled from the model itself. We do not recommend deploying models with different set of available features across different clinical environments.
>
> 6) Could the authors shed light on an example on the MIMIC dataset where their approach produces nonsensical results? This would provide future researchers with insight on how to improve upon your approach.
>
> One limitation of our method is that we evaluate the importance of every observation separately. This can result in imperfect importance assignment in the case of highly correlated signals, because this correlation will be broken. Clinically, it is relevant to derive feature importance over subsets of features. We identify this as important future work in the draft.

---

### Decision · Program_Chairs · 2019-12-19

**Decision:**

Reject

**Comment:**

The paper proposes a definition of and an algorithm for computing the importance
of features in time series classification / regression.
The importance is defined as a finite difference version of standard sensitivity
analysis, where the distribution over finite perturbations is given by a
learned time series model.
The approach is tested on simulated and real-world data sets.

The reviewers note a lack of novelty in the paper and deem the contribution
somewhat incremental, although exposition and experiments have improved compared
to previous versions of the manuscript.

I recommend to reject this paper in its current form, taking into account on the reviews and my own
reading, mostly due t a lack of novelty.
Furthermore, the authors call their method a "counterfactual" approach.
I don't agree with this terminology.
No attempt is made to justify is by linking it to the relevant causal literature
on counterfactuals.
The authors do indeed motivate their algorithm by considering how the classifier
output would change "had an observation been different" (a counterfactual), but
mathematical in their model this the same as asking "what changes if the observation is
different" (interventional query).